# Developing a clinical prediction rule for repeated consultations with functional somatic symptoms in primary care: a cohort study

Gea A Holtman ,[1] Huibert Burger,[1] Robert A Verheij,[2,3] Hans Wouters,[1,4] Marjolein Y Berger,[1] Judith GM Rosmalen,[5] Peter FM Verhaak[1,2]

For numbered affiliations see end of article.

**Correspondence to**
Dr GA Holtman;
g.a.holtman@umcg.nl and
Dr Gea A Holtman;
g.a.holtman@umcg.nl

## ABSTRACT

**Objectives** Patients who present in primary care with chronic functional somatic symptoms (FSS) have reduced quality of life and increased health care costs. Recognising these early is a challenge. The aim is to develop and internally validate a clinical prediction rule for repeated consultations with FSS.

**Design and setting** Records from the longitudinal population-based ('Lifelines') cohort study were linked to electronic health records from general practitioners (GPs).

**Participants** We included patients consulting a GP with FSS within 1 year after baseline assessment in the Lifelines cohort.

**Outcome measures** The outcome is repeated consultations with FSS, defined as ≥3 extra consultations for FSS within 1 year after the first consultation. Multivariable logistic regression, with bootstrapping for internal validation, was used to develop a risk prediction model from 14 literature-based predictors. Model discrimination, calibration and diagnostic accuracy were assessed.

**Results** 18 810 participants were identified by database linkage, of whom 2650 consulted a GP with FSS and 297 (11%) had ≥3 extra consultations. In the final multivariable model, older age, female sex, lack of healthy activity, presence of generalised anxiety disorder and higher number of GP consultations in the last year predicted repeated consultations. Discrimination after internal validation was 0.64 with a calibration slope of 0.95. The positive predictive value of patients with high scores on the model was 0.37 (0.29–0.47).

**Conclusions** Several theoretically suggested predisposing and precipitating predictors, including neuroticism and stressful life events, surprisingly failed to contribute to our final model. Moreover, this model mostly included general predictors of increased risk of repeated consultations among patients with FSS. The model discrimination and positive predictive values were insufficient and preclude clinical implementation.

## Strengths and limitations of this study

► This study offers valuable insights into the predictors that could help general practitioners (GPs) to identify repeated consultations with functional somatic symptoms.

► By linking routine healthcare data from primary care to a large population-based cohort, we could include relevant predictors based on epidemiological and theoretical factors from the literature and this approach may serve to enhance primary care research in the future.

► Each patient had a full follow-up of 1 year.

► Time from baseline assessment of the population-based cohort to first GP consultation varied, however, taking this variance into account did not affect the magnitude of the coefficients of the predictors in a substantial way, nor their selection.

► We did not externally validate the model, however the performance needs to be improved before such research can be considered.

for about one-third of all presentations in primary care,[1 2] clustering as cardiopulmonary, musculoskeletal, gastrointestinal and general somatic symptoms.[3 4] However, these clusters appear to correlate and considered to represent one condition with different manifestations.[5] Most patients with FSS consult a general practitioner (GP) only once, but 10%–30% of cases will become chronic,[6] leading to more diagnostic tests, more referrals, higher healthcare costs and more psychological distress compared with other patients.[7–9] Recognising those patients at risk of developing chronic symptoms and consulted repeatedly the GP could therefore help to target interventions that reduce symptom severity,[10 11] improve quality of life and reduce GP workloads. Ensuring that these patients are identified early is an important challenge facing GPs,[12] and one for which a validated clinical prediction rule may help.

## INTRODUCTION

Functional somatic symptoms (FSS), a synonymous of medically unexplained physical symptoms, represent those that cannot be explained by a physical disease and account

Several factors are known to increase the risk of chronicity of FSS, including predisposing (eg, neuroticism), precipitating (eg, physical and psychosocial stressors) and perpetuating (eg, lack of healthy physical activity) factors.[13–15] Despite being described in the literature,[6] these factors have yet to be combined to predict repeated consultations with FSS in a clinical prediction rule for use in primary care.

In this study, we aimed to develop and internally validate a clinical prediction rule for repeated consultations among patients who consult GPs with FSS.

## METHOD

### Data sources

We linked patient records from the Lifelines Cohort Study ('Lifelines')[16] with those from the Nivel Primary Care Database (NPCD).[17] Dutch law conditionally allows the use of such electronic health records for research purposes. Statistics Netherlands (CBS) then used temporary record identification numbers to link records at an individual level for analysis.

Lifelines is a multidisciplinary prospective population-based cohort study using a three-generation design to examine the health and health-related behaviours of 167 729 people living in the north of the Netherlands.[16] It employs a broad range of investigative procedures to assess key factors that contribute to health and disease in the general population, focusing on multimorbidity and complex genetics. Lifelines was conducted in accordance with the Declaration of Helsinki. All participants signed an informed consent form.

The NPCD contains routinely recorded clinical data from GP consultations with patients, and is considered representative of the Dutch population.[17] The Dutch healthcare system is such that all non-institutionalised members of the population are registered with a general practice, which in turn, serves as a gatekeeping system through which patients must pass to access specialist care via GP referral.[18] In total, 528 general practices participated in 2019, and this study was approved according to the Nivel Governance Code (number NZR0317.033).

For the current study, we included the baseline data of 152 728 adults enrolled in Lifelines between November 2006 and June 2013, and we linked these with the electronic health records of GP consultations for patients aged ≥18 years who consulted one of the 65 general practices in the north of the Netherlands that participated in the NPCD.

### Patient population

We planned to include adults with FSS considered at risk of consulting the GP repeatedly, which we defined as those having a GP consultation for FSS in the year after their baseline assessment for Lifelines. The presence of FSS was assessed based on the International Classification of Primary Care (ICPC) codes that related to the

symptoms that Robbins *et al* described (see online supplemental table 1).[19]

### Outcomes

The primary outcome was repeated consultations with FSS, defined as ≥3 extra GP consultations for one of the defined FSS (see online supplemental table 1) during a year of follow-up after first consulting a GP with that symptom.[19 20] Complete follow-up data were recorded for all GP consultations in electronic health records, and we permitted the FSS to vary between consultations.

### Candidate predictors

We selected 14 predictors based on literature review and expert opinion: age, sex, neuroticism, chronic stress, stressful life events, self-rated health, healthy activity, body mass index (BMI), living alone, higher education, major depressive disorder (MDD), generalised anxiety disorder (GAD), and psychiatric or GP consultations in the 12 months before first consulting with FSS.[6 21] The data for these predictors were derived from the baseline of Lifelines, except for the psychiatric and GP consultations, which were derived from the NPCD.

Neuroticism was evaluated using an abridged version of the Neuroticism Extraversion Openness-Personality Inventory-Revised that included only anger-hostility, self-consciousness, impulsivity and vulnerability, and excluded depression and anxiety (score range, 4–32).[22] Chronic stress was measured with the Long-term Difficulties Inventory (score range, 0–24).[23] The List of Threatening Events was used to assess the occurrence of 12 stressful life events (score range, 0–12).[23 24] Self-rated health was evaluated with the RAND-36 question[25] 'how would you rate your health from 1 (excellent) to 5 (poor).' The Short Questionnaire to Assess Health-Enhancing Physical Activity was used to determine healthy activity behaviour, with a cut-off of 30 min at least 5 days a week indicating healthy activity.[26] Body weight and height were used to calculate BMI (weight (kg)/height (m$^2$)). Higher education was defined as at least secondary vocational education or work-based training. MDD and GAD were assessed by the Mini-International Neuropsychiatric Interview, compatible with International Classification of Disease, Tenth Edition, and the Diagnostic and Statistical Manual of Mental Disorders, Fourth Edition.[27] Psychiatric consultation was defined as patients with a consultation code in the P chapter of the ICPC and GP consultation was defined as the number of total GP consultations in the 12 months before baseline of Lifelines.

### Sample size

We estimated that we required 11 455 participants based on an assumption that 10% of the NPCD cohort would participate in Lifelines and that 75% of these data could be linked (ie, 10%×75%×152 728). Given that the prevalence of repeated consultations with FSS has been reported to be 2.5%, we estimated that 286 of these could

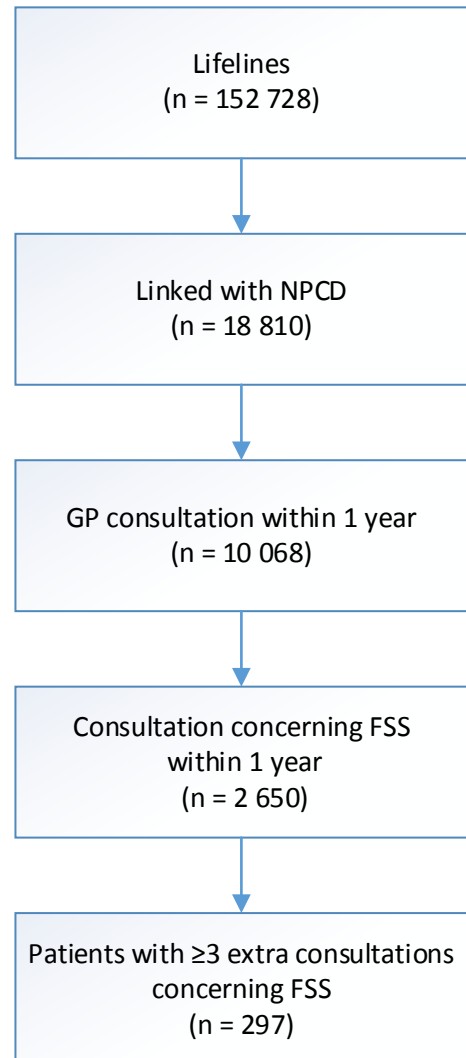

**Figure 1** Patient selection flowchart. 'Extra consultations' (≥3) refers to additional presentations for FSS during a 1-year follow-up period after an initial GP consultation for FSS. GP, general practitioner; FSS, functional somatic symptoms.

be included[20] to achieve an effective sample size of at least 20 outcome events per predictor.[28]

## Missing data

Eleven predictors from Lifelines had missing data, so we evaluated the underlying causes and patterns to assess the conditions for multiple imputation.[29] We checked predictors of missingness and we assumed missing at random (MAR) when patients with missing values were different from patients without missing values with respect to observed variables. When data are MAR, we replaced all missing values by multiple imputation by chained equations, incorporating all variables used in the analyses, including the outcome variable, and all variables that predicted missingness of a certain variable or value. We imputed questionnaire sum scores rather than item scores. Finally, we constructed 20 imputed data sets combined across all data sets, pooled β coefficients and calculated ORs using Rubin's rule.[30]

## Statistical analysis

Repeated consultations with FSS over a 1-year follow-up period was set as the binary outcome variable and associated with potential predictors as independent variables in logistic regression analyses. We performed univariable analyses to calculate unadjusted ORs.

To develop the clinical prediction rule, we initially included all potential predictors in a multivariable logistic regression model, irrespective of their univariable association and refrained from univariable preselection of candidate predictors to prevent model instability.[31] Using backward stepwise selection, we excluded predictors from the model that were not statistically significant according to Akaike's information criterion (ie, p>0.157) in >50% of all imputed data sets.[32] Time in days between baseline assessment of predictors and first consultation differed between participants, so we also evaluated its influence in a separate analysis. We assessed rule performance by its discriminatory power with the C statistic and the calibration slope. We internally validated the model to correct for overoptimism by bootstrapping 250 samples, calculating a shrinkage factor, multiplying the original β coefficients by this factor, and re-estimating the intercepts using the shrunken β coefficients. The β coefficients were translated into a risk score of whole numbers for ease of use by GPs when evaluating the risk of repeated consultations in clinical practice. To that end, each β coefficient was divided by the coefficient closest to zero and then rounded to the nearest integer. The total score for each patient was calculated as the sum of all points for each predictor. We calculated the sensitivity, specificity and positive predictive value of the rule at several thresholds to distinguish high and low risk. Thresholds were chosen arbitrarily based on the sample sizes being adequate in each category and the clinical risk being distinguishable.

All statistical analyses were performed with Stata V.SE15 (StataCorp, College Station, Texas, USA) and R (for bootstrapping). The Transparent Reporting of a multivariable prediction model for Individual Prediction of Diagnosis was used to conduct this study and report its results.[33]

## Patient and public involvement

Lifelines has a participant advisory board of eight active members with different backgrounds since 2016. The concept of this study was discussed during a meeting with this board. All Lifelines participants will receive the results of the study via a newsletter.

## RESULTS
### Study participants

Of the 152 728 Lifelines participants with a baseline assessment, we linked 18 810 (12%) with NPCD data (figure 1). Among these, we included 2650 participants (14% of those linked) attending GP consultations for FSS (ie, the at-risk group), of whom 297 (11%) had ≥3 further consultations for FSS (ie, outcome criterion). The details of the included and

**Table 1**  Characteristics of included and excluded patients

| | Included patients* (n=2650) | | Excluded patients† (n=7418) | |
|---|---|---|---|---|
| | N (%) | Score | n | Score |
| Age, mean years (SD) | 2636 (99) | 45 (14) | 7387 | 45 (14) |
| Female, n (%) | 2650 (100) | 1802 (68) | 7418 | 4577 (62) |
| Neuroticism, median (IQR) | 2248 (85) | 10.1 (9.1–11.3) | 6419 | 9.9 (8.9–11) |
| Chronic stress, median (IQR) | 2465 (93) | 2 (1–4) | 7005 | 2 (1–4) |
| Stressful life events, median (IQR) | 2464 (93) | 1 (0–2) | 7008 | 1 (0–2) |
| Self-rated health, median (IQR) | 2548 (96) | 3 (2–3) | 7228 | 3 (2–3) |
| Healthy activity‡, n (%) | 2274 (86) | 1259 (55) | 6505 | 3589 (55) |
| Body mass index (kg/m$^2$), median (IQR) | 2648 (100) | 26 (23–28) | 7417 | 25 (23–28) |
| Living alone, n (%) | 2522 (95) | 335 (13.3) | 7167 | 904 (12.6) |
| Higher education§, n (%) | 2572 (97) | 1707 (66) | 7215 | 5067 (70) |
| MDD, n (%) | 2555 (96) | 86 (3) | 7248 | 176 (2.4) |
| GAD, n (%) | 2555 (96) | 165 (6) | 7248 | 351 (4.8) |
| Psychiatric consultations last year¶**, n (%) | 2650 (100) | 292 (11) | 7418 | 783 (11) |
| GP consultations last year**, median (IQR) | 2650 (100) | 2 (0–5) | 7418 | 1 (0–3) |

*Included: GP consultations and ≥1 FSS within 1 year after baseline Lifelines assessment.
†Excluded: GP consultations without FSS within 1 year after baseline of Lifelines.
‡Healthy activity, defined as 30 min at least 5 days a week.
§Higher education, defined as at least secondary vocational education or work-based training.
¶Patients with a consultation code in the P chapter of the International Classification of Primary Care.
**Predictors from NPCD. Other predictors are from Lifelines.
GAD, generalised anxiety disorder; GP, general practitioner; MDD, major depressive disorder; NPCD, Nivel Primary Care Database.

excluded patients are summarised in table 1, showing that the groups were broadly comparable. Notably, 24% of participants had a missing value and 3% had missing values for >4 predictors. The participants with missing values were slightly older and less active, and they less often had completed higher education (see online supplemental table 2).

### Clinical prediction rule

Univariable associations of the potential predictors for repeated consultations with FSS are listed in table 2. In the final multivariable model, the following five predictors were selected based on increasing the risk of repeated consultations: higher age, female sex, lack of healthy activity, presence of GAD and having had more GP consultations in the year before first consulting with FSS (table 3). Adjustment for time from baseline to first consultation did not affect the magnitude of the coefficients of the predictors in a substantial way, nor their selection. The shrinkage factor of 0.95 showed limited model overfitting and was applied to adjust predictor coefficients in the final model. Likewise, the C statistic (area under the curve) of 0.65 (95% CI, 0.62 to 0.69) was corrected to 0.64 (95% CI, 0.61 to 0.68). Agreement between the observed and predicted proportion of events showed adequate calibration (see online supplemental figure 1).

The final model could calculate the absolute predicted individual risk of repeated consultations with FSS (see online supplemental figure 2). For a risk score ≥100, the positive predictive value of repeated consultations was 0.37 (95% CI, 0.29 to 0.47) (tables 4 and 5). However, when increasing the cut-off from 25 to 100, the sensitivity decreased from 0.87 (95% CI, 0.83 to 0.91) to 0.13 (95% CI, 0.10 to 0.17) and the specificity increased from 0.23 (95% CI, 0.22 to 0.25) to 0.97 (95% CI, 0.97 to 0.98).

### DISCUSSION
### Summary

We developed and internally validated a clinical prediction rule to identify patients at high risk of repeated consultations with FSS. This was based on five factors that are readily available in primary care: age, sex, activity levels, GAD diagnosis and number of consultations. However, despite being well calibrated, the prediction rule showed poor discrimination. Nevertheless, if patients scored ≥100, the risk of repeated consultations with FSS increased to 37% from the baseline value of 11%.

### Strengths and limitations

The study benefited from the use of a rich data set established by linking routine electronic health record data from primary care to a large population-based

**Table 2** Univariable analysis of predictors for repeated consultations with FSS

| Variable | Repeated consultations* OR (95% CI) |
|---|---|
| Age | 1.01 (1.00 to 1.02) |
| Sex (male) | 0.69 (0.52 to 0.91) |
| Neuroticism | 1.08 (0.99 to 1.17) |
| Chronic stress | 1.04 (0.99 to 1.09) |
| Stressful life events | 1.08 (0.99 to 1.18) |
| Self-rated health | 1.41 (1.20 to 1.66) |
| Healthy activity† | 0.70 (0.53 to 0.91) |
| Body mass index (kg/m$^2$) | 1.04 (1.01 to 1.07) |
| Living alone | 0.91 (0.62 to 1.32) |
| Higher education‡ | 0.75 (0.58 to 0.96) |
| MDD | 1.53 (0.85 to 2.73) |
| GAD | 2.15 (1.45 to 3.19) |
| Psychiatric consultations last year§¶ | 1.17 (1.04 to 1.33) |
| GP consultations last year¶ | 1.12 (1.09 to 1.15) |

*Outcome, ≥3 extra FSS consultations during a 1-year follow-up period (n=297).
†Healthy activity, defined as 30 min at least 5 days a week.
‡Higher education, defined as at least secondary vocational education or work-based training.
§Number of consultations concerning International Classification of Primary Care codes in the P chapter.
¶Predictors are from NPCD and are continuous. Other predictors are from the Lifelines database.
FSS, functional somatic symptoms; GAD, generalised anxiety disorder; GP, general practitioner; MDD, major depressive disorder; NPCD, Nivel Primary Care Database.

**Table 4** Risk of repeated consultations with FSS by different cut-off scores

| Cut-off score | n | Outcome | Observed risk | Predicted risk |
|---|---|---|---|---|
| <25 | 585 | 38 | 0.07 | 0.06 |
| 25–49 | 1009 | 82 | 0.08 | 0.08 |
| 50–99 | 952 | 138 | 0.15 | 0.14 |
| ≥100 | 104 | 39 | 0.38 | 0.37 |

The risk score was calculated by multiplying each risk score by the predictor value, with the total score ranging from –21 to 301 for all included patients (for example, –21 represents the following patient: 18 years (18), man (–15), healthy activity (–24), no GAD (0), no GP consultation last year (0) (=18–15–24+0+0=–21); and 301: 63 years (63), woman (0), lack of healthy activity (0), presence of GAD (28), 42 GP consultations last year (210) (=63+0+0+28+210=301)).
FSS, functional somatic symptoms; GAD, generalised anxiety disorder; GP, general practitioner.

cohort. We effectively linked 18 810 patients (12%) from Lifelines who had at least one GP consultation, and we could include predictors based on epidemiological and theoretical factors from the literature, such as neuroticism and stressful life events.[21] The data linkage approach that we adopted may serve to enhance primary care research in the future. Each patient also had follow-up data for a full year, and although the time from baseline assessment in

Lifelines to first GP consultation varied because of the dynamic nature of the NPCD cohort, this did not affect the results. Another strength is that we included 21 events per variable, resulting in minimal overfitting with a shrinkage factor of 0.95. An advantage of using dichotomous over continuous outcomes is that clinical interpretation is more straightforward. Although it is problematic that we did not externally validate the model, we contend that the model's performance will need to be improved before such research can be considered.

Our model predicts the risk of having ≥3 extra consultations for FSS. A developing underlying somatic disease could be suggested to ultimately explain some of these symptoms, however, a meta-analysis suggested that this risk is very low, reporting only 0.5% new diagnoses in follow-up studies of FSS.[34] FSS should not be confused with predicting a somatic symptom disorder or functional somatic syndrome, not least because we could not determine these diagnoses with the available data. In addition, about 80% of patients with functional somatic syndrome will be missed using GP medical files.[35] A disadvantage of

**Table 3** Final multivariable analysis for repeated consultations with FSS

| Predictors | OR (95% CI) | P value | Coefficient | Adjusted coefficient | Risk score |
|---|---|---|---|---|---|
| Constant | 0.05 (0.03 to 0.08) | 0.000 | –2.95 | –3.80 | |
| Age | 1.02 (1.01 to 1.03) | 0.000 | 0.02 | 0.02 | 1 |
| Sex (male) | 0.75 (0.56 to 0.99) | 0.042 | –0.30 | –0.29 | –15 |
| Healthy activity* | 0.60 (0.45 to 0.80) | 0.001 | –0.51 | –0.48 | –24 |
| GAD | 1.79 (1.17 to 2.74) | 0.008 | 0.58 | 0.56 | 28 |
| GP consultations last year | 1.10 (1.07 to 1.14) | 0.000 | 0.10 | 0.10 | 5 |

Shrinkage factor 0.95; predictors selected if p<0.157.
*Healthy activity, defined as 30 min at least 5 days a week.
FSS, functional somatic symptoms; GAD, generalised anxiety disorder; GP, general practitioner.

**Table 5** Diagnostic accuracy of the risk score for repeated consultations with FSS

| Cut-off score | n | Sensitivity (95% CI) | Specificity (95% CI) | PPV (95% CI) | NPV (95% CI) |
| --- | --- | --- | --- | --- | --- |
| ≥25 | 2065 | 0.87 (0.83 to 0.91) | 0.23 (0.22 to 0.25) | 0.13 (0.11 to 0.14) | 0.93 (0.91 to 0.95) |
| ≥50 | 1057 | 0.59 (0.54 to 0.65) | 0.63 (0.61 to 0.64) | 0.17 (0.15 to 0.19) | 0.92 (0.91 to 0.94) |
| ≥100 | 104 | 0.13 (0.10 to 0.17) | 0.97 (0.97 to 0.98) | 0.37 (0.29 to 0.47) | 0.91 (0.90 to 0.92) |

FSS, functional somatic symptoms; NPV, negative predictive value; PPV, positive predictive value.

our outcome measure is that patients with FSS may also have consulted other healthcare professionals (eg, physiotherapist), so these cases may have been missed. Therefore, the interpretation of our model is only applicable for GP consultations. We chose to use a follow-up of 1 year as this is often used in previous studies,[20 36] however persistent frequent attenders in primary care have more often FSS.[37] Therefore, a clinical prediction rule for repeated consultations with FSS during a longer follow-up might perform better. To avoid confusion and misunderstanding, we used the more neutral outcome of repeated consultations because our data did not allow for the identification of frequent attenders as defined in present day literature. The latter requires a comprehensive description of the contacts counted, for example, how many were out-of-hours contacts, and how many were administrative or preventive consultations.[36] As we did not want to include too many predictors per variable to prevent overfitting, we a priori choose which predictors were relevant and feasible to use in a primary care setting. By this arbitrary selection, we may have missed relevant predictors (e.g. panic disorder and number of physical symptoms) that could have improved the performance of our prediction rule.[6 38] Our approach to identify the at-risk population first may explain the contrast with existing data. For example, we showed that 11% of patients presenting with FSS ultimately had ≥4 consultations for these symptoms, whereas previous research has shown a rate of 2.5% among all patients with GP consultations.[20]

### Comparison with other studies

We are aware of no other clinical prediction rules for repeated GP consultations with FSS. It should be emphasised that such a model cannot be considered synonymous with explaining the cause.[39] However, we found three studies that developed models for persistent FSS by combining predictors using a backward or forward selection procedure. We limit our discussion to the three studies that developed a clinical prediction rule.

The first study used information from GP letters to medical specialists for patients who were referred with FSS.[40] In their clinical prediction rule, female sex, referral symptom group, lack of somatic comorbidity, lack of abnormal physical findings, history of psychiatric diagnosis or treatment, and referral letter written in illness terminology were all shown to be predictors

for FSS. This model had a higher area under the curve (0.82) than ours (0.64) and was developed for patients consulting internists. However, the GP referral letters included relevant predictors that helped to identify FSS, and although the population was more selected than ours, the results show that data collected in primary care can be suitable predictors.

The second study showed that the use of routine healthcare could include relevant predictors and developed a clinical prediction rule that potentially could be used to identify patients at risk for persistent FSS from routine primary care medical records.[41] The model had an area under the curve of 0.70 and the most important discriminative variable for persistent FSS was number of episodes. Just like our model they also included the predictors age, sex and number of contacts.

The third study developed a model for symptom severity and for both physical and mental functioning during a 2-year follow-up period among patients with persistent FSS.[21] They predicted severe courses by physical comorbidity, higher baseline severity and longer physical symptom duration, anxiety, catastrophising cognitions, embarrassment and neuroticism, as well as fear avoidance, avoidance or resting behaviour. By contrast, they predicted favourable courses based on limited alcohol use, higher education, higher baseline physical and mental functioning, symptom focusing, damage cognitions and extraversion. Although we also identified anxiety as a predictor, we did not find the same for neuroticism or higher education. Also contrasting with our data, as well as that of others,[42 43] they did not show that female sex was a predictor. Unfortunately, we could not include predictors of illness behaviour because these were not evaluated in Lifelines. Indeed, the Symptoms Checklist 90 questionnaire had more than 50% missing values during baseline evaluation in Lifelines, so we excluded these data.[16] The differences in identified predictors may be explained by different study populations, predictor selection criteria or outcomes.

### Implications for research and practice

To our surprise, several theoretically suggested predisposing and precipitating predictors, including neuroticism and stressful life events, failed to contribute to the final prediction model. Instead, this model included mostly general predictors that provide little

additional information to help GPs recognise patients at risk of consulting repeatedly with FSS, and it not only has poor discrimination and positive predictive value but also lacks external validation. Therefore, at present, we cannot recommend the score for clinical use. Nevertheless, our findings indicate that GPs might expect chronicity when older women with low activity levels and anxiety symptoms present with FSS. These require extra vigilance and may benefit from early intervention with self-help advice.[11] Some predictors identified in earlier studies, such as female sex and anxiety, could be potential factors in future clinical prediction rules designed to help GPs recognise patients at risk of consulting the GP repeatedly.

**Author affiliations**
[1]Department of General Practice and Elderly Care Medicine, University of Groningen, University Medical Center Groningen, Groningen, The Netherlands
[2]NIVEL, Netherlands Institute of Health Services Research, Utrecht, The Netherlands
[3]Tranzo Scientific Center for Care and Welfare, Tilburg School of Social and Behavioral Sciences, Tilburg University, Tilburg, The Netherlands
[4]General Practitioners Research Institute, Groningen, The Netherlands
[5]Interdisciplinary Center Psychopathology and Emotion Regulation, University of Groningen, University Medical Center Groningen, Groningen, The Netherlands

**Correction notice** This article has been corrected since it first published. The provenance and peer review statement has been included.

**Acknowledgements** We thank Dr Robert Sykes (www.doctored.org.uk) for providing editorial services.

**Contributors** GAH, HB and PFMV—study concept and design. PFMV and RAV—acquisition of data. GAH—analysis of data. GAH, HB, RAV, HW, MYB, JGMR and PFMV—interpretation of data. GAH, HB and PFMV—drafting the work. The final manuscript was critically revised and approved by all authors.

**Funding** This study was funded by a grant from the Healthy Ageing Pilot of the University Medical Center Groningen (CD017.0002/2017-1/296). The Lifelines initiative has been made possible by support from the Dutch Ministry of Health, Welfare and Sport; the Dutch Ministry of Economic Affairs; University Medical Center Groningen; Groningen University; and the Provinces in the north of the Netherlands (Drenthe, Friesland, Groningen).

**Competing interests** None declared.

**Patient consent for publication** Not required.

**Ethics approval** The medical ethics committee of the University Medical Centre Groningen approved the Lifelines Study (2007/152). Dutch law conditionally allows the use of electronic health records for research purposes.

**Provenance and peer review** Not commissioned; externally peer reviewed.

**Data availability statement** Data were obtained from a third party and are not publicly available. The data used are part of two ongoing databases: Lifelines Cohort Study and Nivel Primary Care Database. The data are not available. The corresponding author can be contacted for details.

**ORCID iD**
Gea A Holtman http://orcid.org/0000-0001-6579-767X

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
