## [Reviewer comments · BMJ Open]

ARTICLE DETAILS

TITLE (PROVISIONAL)	Developing a clinical prediction rule for repeated consultations with functional somatic symptoms in primary care, a cohort study
AUTHORS	Holtman, G.; Burger, H; Verheij, Robert A; Wouters, Hans; Berger, Marjolein; Rosmalen, Judith; Verhaak, Peter FM

VERSION 1 – REVIEW

REVIEWER	Frans T. Smits dep general practice UMC Amsterdam The Netherlands
REVIEW RETURNED	08-Jun-2020

GENERAL COMMENTS	Review of bmjopen-2020-040730: "Developing a clinical prediction rule for frequent attenders with functional symptoms in primary care". Gea Holtman et coll. developed and internally validated a clinical prediction rule for repeated consultations for functional complaints after an initial presentation for this same problem. Physical complaints, not explained by physical illness, are indeed accountable for a large part of the work of GP's and diminish the quality of life of many of our patients. Therefore, prediction of repeated use of healthcare for these reasons can be useful. The linkage of NPCD to the large Lifelines data base is certainly interesting. The authors clearly describe the problem they want to address, the methodology and the conclusions. Main remarks: 1. The authors use the term 'frequent attender' rather different from the international literature. Nowadays there is agreement about the definition of Frequent attenders (FA) in primary care and most authors use not a threshold definition, but a proportional definition of FA per sex and age group (upper 10%).¹ Using FA for repeated consultations after an initial presentation for a functional complaint is confusing and leads to, in my opinion, unnecessary misunderstandings. I would strongly suggest to change this, throughout all the article, in something like 'repeated consultations'.2. The authors use 'functional symptoms' for complaints that cannot be explained by a physical disease. This is also the definition of the, in general practice, frequently used term 'Medically unexplained physical symptoms (MUPS)' or the psychiatric diagnosis 'somatic symptom disorder'. The Robbins list is also used to define MUPS in primary care.² They don't discuss why they have chosen for 'functional symptoms', why they don't use the commonly used terms MUPS or somatic symptom disorder
---

and why they insufficiently use, describe and discuss the large literature about MUPS and somatic symptom disorder. (E.g. Madelon den Boeft et coll. already developed in her thesis a risk assessment model for patients with persistent medically unexplained physical symptoms.³)

3. I don't think one can speak of a 'functional symptom' when a patient visits a GP only once for a provisional, 'symptom diagnosis' conform the Robbins list. Often this will be a temporary diagnosis pending a final, somatic or psychiatric, diagnosis. This may have biased your results and perhaps inclusion of patients after e.g. 2 consultations with the same functional symptom could diminish this bias and change your prediction model.

4. In the literature describing persistent FA was found that panic disorder (more than GAD), negative life events in the previous year, illness behavior and lack of mastery were associated with persistent frequent attendance.⁴ I miss panic disorder and mastery as a predictive factor in your study. Of course, I don't know whether you have data about panic disorder and whether it is possible to include these possible predictors in your model? It would be interesting and useful to compare and discuss the presented risk model with risk models for (persistent) FA. Did you consider to include (the number of) MUPS-episodes and psychiatric medication (tranquilizers; antidepressants) as a candidate predictor?

5. You have chosen a rather short follow-up period of 1 year. Knowing that persistent frequent users (FA during 2-3 years) have more MUPS², it would be interesting to investigate whether a prediction rule for persistent functional complaints might perform better.

Detailed remarks:

6. Title and article: Please change 'frequent attender' (see remark 1).
7. P2, l26: '...defined as >3 extra consultations for a functional complaint within 1 year after the first consultation'.
8. P2, l34: Please change 'attended frequently' in 'attended > 3 times'.
9. P2, l48: 'however'? 'Moreover'?
10. P6, l15: I miss the for functional symptoms often used Illness Attitude Scale.
11. P6, l20: You are unclear whether ANY psychiatric OR any GP consultation in the 12 months before inclusion was sufficient (also for a somatic diagnosis) or that you used a continuous variable. Please explain more clearly. Interesting candidate predictors may have been frequent attendance (upper 10% for age and sex) to the GP in 1 or 2 years before inclusion, the N of (MUPS)episodes and the N of (P) medication.^{5;3}
12. P7, l30: As far as I can see the statistical analysis is OK, but I am insufficiently capable to fully judge the soundness of this analysis.
13. P9, l34: Please also mention in your discussion the items of my main remarks.
14. P10, l47: Other authors constructed prediction rules for MUPS (Smith, Morriss, den Boeft). Please discuss remark 4.
15. Table 1: I don't understand why you don't have the age of all included and excluded patients. Please explain all abbreviations in the legend (SD, IQR etc.)

	1 Vedsted P, Christensen MB. Frequent attenders in general practice care: a literature review with special reference to methodological considerations. Public Health 2005; 119: 118–37. 2 Smits FT, Brouwer HJ, Ter Riet G, Van Weert HCP. Epidemiology of frequent attenders: A 3-year historic cohort study comparing attendance, morbidity and prescriptions of one-year and persistent frequent attenders. BMC Public Health 2009; 9. DOI:10.1186/1471-2458-9-36. 3 den Boeft M. Risk assessment models for patients with persistent medically unexplained physical symptoms in primary care using electronic medical records. 2016. https://research.vu.nl/en/publications/medically-unexplained-physical-symptoms-in-primary-care-identific. 4 Smits FT, Brouwer HJ, Zwinderman AH, et al. Why do they keep coming back? Psychosocial etiology of persistence of frequent attendance in primary care: A prospective cohort study. J Psychosom Res 2014; 77: 492–503. 5 Smith RC, Gardiner JC, Armatti S, et al. Screening for high utilizing somatizing patients using a prediction rule derived from the management information system of an HMO: A preliminary study. Med Care 2001; 39: 968–78.
--	--

REVIEWER	VERA MARIA VIEIRA PANIZ Universidade do Vale do Rio dos Sinos São Leopoldo, RS, BRAZIL
REVIEW RETURNED	07-Jul-2020

GENERAL COMMENTS	General comments: The manuscript presents a complex analysis by linking routine electronic health record data from primary care to a large population-based cohort. The aim of the study is to develop and internally validate a clinical prediction rule for frequent attenders with functional symptoms. According to the literature the best design to address prognostic questions is a cohort study. However, prognosis studies usually assess which baseline characteristics of patients with functional or physical symptoms predict symptom severity over a longer follow-up period, about two years. In the present study, each patient had a complete follow-up of 1 year, but the time between the initial evaluation of the population-based cohort and the first GP visit varied. The authors mention that this did not affect the results. Authors need to clarify this. The study has strengths and limitations that impact results and need to be detailed. A weakness of the research is the high number of missings that is not shown in tables. Another disadvantage concerns the classification error that may have occurred as a result of the outcome if we consider that patients with functional symptoms may have consulted other health professionals during the period under investigation. This must be discussed. Although the final model obtained has not confirmed the previous hypotheses and its potential to contribute to the clinician is still limited by the low positive predictive value, and the lack of external validation of the models presented, the manuscript innovates in the methodology used and shows that data collected in primary care can be adequate predictors.
--

	The data linkage approach adopted may serve to enhance primary care research in the future. Other issues to be addressed: 1- Candidate predictors: Quality of life should be renamed to self-rated health. Report the origin of the information used to calculate the Body Mass Index. 2- Missing data: The method of data imputation must be detailed and justified. 3- Statistical analysis: The time between baseline assessment of predictors and first consultation differed between participants, and their influence was evaluated in a separate analysis. Provide a defense of this. The total score for each patient was calculated as the sum of all points for each predictor. Detail the total score ranging from -21 to 301 according to the predictors. 4- Results: 24% of participants had a missing value and 3% had missing values for >4 predictors. Detail in table. Only five predictive factors out of the 14 included in the analysis remained in the final predictive model. What do the authors attribute this to? Standardize the predictor category. Male or female? Lack of healthy activity or healthy activity? Risk or protection factor? 5- Tables and Figure Table 1. Characteristics of included and excluded patient should be deleted. The Figure 1 is sufficient. The Table 1 should show the predictors characteristics of study population at baseline, overall group and missing (N%). Table 2. should contain the univariate analysis presented as Supplementary Table 3. Univariable analysis of predictors for frequent attendance with functional symptoms Figure 2. Relation between the total risk score and the predicted risk of frequent attendance with functional symptoms as supplementary material.
--	---

VERSION 1 – AUTHOR RESPONSE

Reviewer: 1 Frans T. Smits

Gea Holtman et coll. developed and internally validated a clinical prediction rule for repeated consultations for functional complaints after an initial presentation for this same problem. Physical complaints, not explained by physical illness, are indeed accountable for a large part of the work of GP's and diminish the quality of life of many of our patients. Therefore, prediction of repeated use of healthcare for these reasons can be useful. The linkage of NPCD to the large Lifelines data base is certainly interesting. The authors clearly describe the problem they want to address, the methodology and the conclusions.

We would like to thank the reviewer for the positive comments about our study. We are happy to hear that the problem, methodology and conclusions are clearly described.

Main remarks:

1. The authors use the term 'frequent attender' rather different from the international literature. Nowadays there is agreement about the definition of Frequent attenders (FA) in primary care and most authors use not a threshold definition, but a proportional definition of FA per sex and age group (upper 10%).¹

Using FA for repeated consultations after an initial presentation for a functional complaint is confusing and leads to, in my opinion, unnecessary misunderstandings. I would strongly suggest to change this, throughout all the article, in something like 'repeated consultations'.

Thank you for the suggestion. Unlike Vedsted 2005 et al and Smits 2009 et al., we studied frequent attendance for specified 'functional symptoms'. However, for clarity we changed 'frequent attender' to 'repeated consultations' as the outcome measure throughout the manuscript. In addition, we explained the difference between frequent attender and repeated consultations in the discussion: 'To avoid confusing and misunderstanding, we used the more neutral outcome of repeated consultations because our data did not allow for the identification of frequent attenders as defined in present day literature. The latter requires a comprehensive description of the contacts counted, e.g. how many were out-of-hours contacts, and how many were administrative or preventive consultations (Vedsted et al. 2005).' (page 11)

2. The authors use 'functional symptoms' for complaints that cannot be explained by a physical disease. This is also the definition of the, in general practice, frequently used term 'Medically unexplained physical symptoms (MUPS)' or the psychiatric diagnosis 'somatic symptom disorder'. The Robbins list is also used to define MUPS in primary care.² They don't discuss why they have chosen for 'functional symptoms', why they don't use the commonly used terms MUPS or somatic symptom disorder and why they insufficiently use, describe and discuss the large literature about MUPS and somatic symptom disorder.

(E.g. Madelon den Boeft et coll. already developed in her thesis a risk assessment model for patients with persistent medically unexplained physical symptoms.³)

With our model, we aimed to predict the risk of having repeated consultations with functional symptoms and not to predict a somatic symptom disorder or MUPS as explained in our discussion (see below). Therefore, we merely refer to 'symptoms'. To make this distinction clear we used the term 'functional symptoms'. In addition, the term 'functional symptoms' was also used in the Robbins paper. In the discussion, we stated: 'Our model predicts the risk of having ≥ 3 extra consultations for functional symptoms. However, this should not be confused with predicting a somatic symptom disorder or functional somatic syndrome, not least because we could not determine these diagnoses with the available data. In addition, about 80% of patients with functional somatic syndrome will be missed using GP medical files (den Boeft et al. 2014).' (page 10)

We did not discuss the paper of Smits et al. 2009 extensively, as the outcome used in our study differs and the authors did not develop a clinical prediction model. However, we discussed this paper briefly when discussing the duration of our follow-up (see point 5).

We missed the papers by den Boeft et al., because one paper which was interesting for our discussion is not yet published (only in thesis). We now added the results of chapter 3 (Risk assessment models for patients with persistent medically unexplained physical symptoms in primary care using electronic medical records) to our discussion: Another study that showed that routine

health care could include relevant predictors, developed a clinical prediction rule that potentially could be used to identify patients at risk for persistent functional somatic symptoms from routine primary care medical records (den Boeft thesis). The model had an area under the curve of 0.70 and the most important discriminative variable for persistent functional somatic symptoms was number of episodes. Just like our model they also included the predictors age, sex, and number of contacts. Other selected predictors were physiotherapy, number of referrals, some medications, number of prescription products with regards to medication, number of trade products with regards to medication, free text questions, and number of laboratory results. (page 12)

In addition, we added the following sentence to the discussion based on another paper of den Boeft: 'In addition, about 80% of patients with functional somatic syndrome will be missed using GP medical files (den Boeft et al. 2014).' (page 10)

3. I don't think one can speak of a 'functional symptom' when a patient visits a GP only once for a provisional 'symptom diagnosis' conform the Robbins list. Often this will be a temporary diagnosis pending a final, somatic or psychiatric, diagnosis. This may have biased your results and perhaps inclusion of patients after e.g. 2 consultations with the same functional symptom could diminish this bias and change your prediction model.

We agree with the reviewer that patients consulting once cannot be regarded as having a provisional 'symptom diagnosis' according to Robbins. However, for the GP it is a challenge to differentiate as early as possible between patients who will become chronic and consult repeatedly and who will not. Therefore, we reasoned that it would be most helpful to recognize patients at risk when they attend with a functional symptom for the first time. We therefore think that this will not bias our result.

4. In the literature describing persistent FA was found that panic disorder (more than GAD), negative life events in the previous year, illness behavior and lack of mastery were associated with persistent frequent attendance.⁴ I miss panic disorder and mastery as a predictive factor in your study. Of course, I don't know whether you have data about panic disorder and whether it is possible to include these possible predictors in your model? It would be interesting and useful to compare and discuss the presented risk model with risk models for (persistent) FA. Did you consider to include (the number of) MUPS-episodes and psychiatric medication (tranquilizers; antidepressants) as a candidate predictor?

The predictors' panic disorder and lack of mastery were not evaluated in Lifelines and were not available in the NPCD data. However, we added the predictor generalized anxiety disorder (GAD), which increased the risk of repeated consultations of functional symptoms. We added the following sentence to the discussion: 'However, there are other relevant potential predictors that were unfortunately not evaluated in Lifelines and were not requested from the NPCD data, such as panic disorder, lack of mastery, medically unexplained physical symptoms episodes, and psychiatric medication (tranquilizers, antidepressants)'. (page 10)

In our discussion, we mention studies on prediction rules for persistent functional symptoms and make a brief comparison with our results. We did not discuss all studies evaluating associations as there are many studies with varying results and we therefore limit our discussion to studies that developed a clinical prediction rule.

5. You have chosen a rather short follow-up period of 1 year. Knowing that persistent frequent users (FA during 2-3 years) have more MUPS2, it would be interesting to investigate whether a prediction rule for persistent functional complaints might perform better.

We agree with the reviewer that it is interesting to evaluate whether a prediction rule for persistent functional complaints during 2 years would perform better. However, many papers used a follow-up of 1-year. Therefore, we conformed to this duration which allows valid comparison of results. Further, there is no generally accepted definition of frequent attendances in general practice (Vedsted et al. 2005). In addition, we believe it is clinically more relevant to know if someone will consult repeatedly in the coming year instead of the coming 2 or 3 years, because it is important to recognize these patients early. Finally, a longer follow-up period would also imply more loss to follow-up (patients moving house, changing general practitioners, and discontinued practices). Therefore, we added this as a limitation to the discussion: 'We chose to use a follow-up of 1 year as this is often used in previous studies (Verhaak et al 2014, Vedsted et al 2005), however persistent frequent attenders in primary care have more often functional somatic symptoms (Smits et al. 2009). Therefore, a clinical prediction rule for repeated consultations with functional symptoms during a longer follow-up might perform better.'

Detailed remarks:

Thank you for all detailed remarks. Below we provide a response to a selection of remarks and the other ones were adjusted in the manuscript.

6. Title and article: Please change 'frequent attender' (see remark 1).
7. P2, I26: '...defined as >3 extra consultations for a functional complaint within 1 year after the first consultation'.
8. P2, I34: Please change 'attended frequently' in 'attended > 3 times'.
9. P2, I48: 'however'? 'Moreover'?
10. P6, I15: I miss the for functional symptoms often used Illness Attitude Scale.

Regarding remark 10, we addressed this in the discussion that this was not evaluated in Lifelines: 'Unfortunately, we could not include predictors of illness cognitions or attitude because these were not evaluated in Lifelines. Indeed, the Symptoms Checklist 90 questionnaire had more than 50% missing values during baseline evaluation in Lifelines, so we excluded these data.'

11. P6, I20: You are unclear whether ANY psychiatric OR any GP consultation in the 12 months before inclusion was sufficient (also for a somatic diagnosis) or that you used a continuous variable. Please explain more clearly. Interesting candidate predictors may have been frequent attendance (upper 10% for age and sex) to the GP in 1 or 2 years before inclusion, the N of (MUPS)episodes and the N of (P) medication.5;3

To make our definition more clear, we added this to the method section: 'Psychiatric consultations was defined as patients with a consultation code in the P chapter of the ICPC and GP consultations was defined as the number of total GP consultations in the 12 months before baseline of Lifelines.'

12. P7, I30: As far as I can see the statistical analysis is OK, but I am insufficiently capable to fully judge the soundness of this analysis.
13. P9, I34: Please also mention in your discussion the items of my main remarks.
14. P10, I47: Other authors constructed prediction rules for MUPS (Smith, Morriss, den Boeft). Please discuss remark 4.
15. Table 1: I don't understand why you don't have the age of all included and excluded patients.

Please explain all abbreviations in the legend (SD, IQR etc.)

- 1 Vedsted P, Christensen MB. Frequent attenders in general practice care: a literature review with special reference to methodological considerations. *Public Health* 2005; 119: 118–37 PubMed .
- 2 Smits FT, Brouwer HJ, Ter Riet G, Van Weert HCP. Epidemiology of frequent attenders: A 3-year historic cohort study comparing attendance, morbidity and prescriptions of one-year and persistent frequent attenders. *BMC Public Health* 2009; 9. DOI:10.1186/1471-2458-9-36.
- 3 den Boeft M. Risk assessment models for patients with persistent medically unexplained physical symptoms in primary care using electronic medical records. 2016. <https://eur03.safelinks.protection.outlook.com/?url=https%3A%2F%2Fresearch.vu.nl%2Fen%2Fpublications%2Fmedically-unexplained-physical-symptoms-in-primary-care-identific&data=02%7C01%7Cg.a.holtman%40umcg.nl%7C4cd120bc30f747079ca208d822661069%7C335122f9d4f44d67a2fccd6dc20dde70%7C0%7C0%7C637297169046242814&sdata=7MQiO%2FxAIjINbRV391%2Fqv%2FEsO4GnOz%2Bo9RChJr%2BayY4%3D&reserved=0>.
- 4 Smits FT, Brouwer HJ, Zwinderman AH, et al. Why do they keep coming back? Psychosocial etiology of persistence of frequent attendance in primary care: A prospective cohort study. *J Psychosom Res* 2014; 77: 492–503 PubMed .
- 5 Smith RC, Gardiner JC, Armatti S, et al. Screening for high utilizing somatizing patients using a prediction rule derived from the management information system of an HMO: A preliminary study. *Med Care* 2001; 39: 968–78 PubMed .

Reviewer: 2 – Vera Maria Vieira Paniz

The manuscript presents a complex analysis by linking routine electronic health record data from primary care to a large population-based cohort. The aim of the study is to develop and internally validate a clinical prediction rule for frequent attenders with functional symptoms.

According to the literature the best design to address prognostic questions is a cohort study. However, prognosis studies usually assess which baseline characteristics of patients with functional or physical symptoms predict symptom severity over a longer follow-up period, about two years.

We think this is a good suggestion. As we responded to reviewer 1 point 5, we chose for 1-year follow-up as this is often used in previous papers and we believe it is more clinically relevant. In addition, we do not predict symptom severity, but repeated consultations. We added this suggestion to the limitation section of the discussion (see reviewer 1, point 5).

In the present study, each patient had a complete follow-up of 1 year, but the time between the initial evaluation of the population-based cohort and the first GP visit varied. The authors mention that this did not affect the results. Authors need to clarify this.

Adding a variable for the time in days between initial evaluation of the population-based cohort and first GP visit to the model, this did not influence the selection nor the coefficients of the predictors. To clarify this, we adjusted the following sentence: ‘Time from baseline assessment of the population-based cohort to first GP consultation varied, however, taking this variance into account did not affect the magnitude of the coefficients of the predictors in a substantial way, nor their selection.’ (page 3)

The study has strengths and limitations that impact results and need to be detailed. A weakness of the research is the high number of missings that is not shown in tables.

We added the % of missing of each variable in Table 1 and in addition, we presented in the supplementary table the patients with and without missing values.

Another disadvantage concerns the classification error that may have occurred as a result of the outcome if we consider that patients with functional symptoms may have consulted other health professionals during the period under investigation. This must be discussed.

We already addressed this point in the discussion: 'A disadvantage of our outcome measure is that patients with functional symptoms may also have consulted other health care professionals (e.g., physiotherapist), so these cases may have been missed.' Nevertheless we now added: 'Therefore, the interpretation of our model is only applicable for GP consultations.' (page 10)

Although the final model obtained has not confirmed the previous hypotheses and its potential to contribute to the clinician is still limited by the low positive predictive value, and the lack of external validation of the models presented, the manuscript innovates in the methodology used and shows that data collected in primary care can be adequate predictors. The data linkage approach adopted may serve to enhance primary care research in the future.

Thank you for the positive remarks. We agree with the reviewer that although we did not find a model that could be used in clinical practice, the innovative methodology could serve as an example to enhance primary care research in the future.

Other issues to be addressed:

1- Candidate predictors: Quality of life should be renamed to self-rated health. Report the origin of the information used to calculate the Body Mass Index.

We changed the terms 'quality of life' to 'self-rated health' throughout the manuscript. We added information on how BMI was calculated to the methods: Body weight and height were used to calculate BMI (weight (kg)/height (m²)) (page 6).

2- Missing data: The method of data imputation must be detailed and justified. We added more detail to the missing data section of the methods: 'Eleven predictors from Lifelines had missing data, so we evaluated the underlying causes and patterns to assess the conditions for multiple imputation.²⁹ We checked predictors of missingness and we assumed missing at random (MAR) when patients with missing values were different from patients without missing values with respect to observed variables. When data is MAR, we replaced all missing values by multiple imputation by chained equations (MICE), incorporating all variables used in the analyses, including the outcome variable, and all variables that predicted missingness of a certain variable or value. We imputed questionnaire sum scores rather than item scores. Finally, we constructed 20 imputed datasets combined across all datasets, pooled β coefficients, and calculated odds ratios using Rubin's rule.³⁰' (page 7)

3- Statistical analysis: The time between baseline assessment of predictors and first consultation differed between participants, and their influence was evaluated in a separate analysis. Provide a defense of this.

When we added a variable of the time in days between initial evaluation of the population-based cohort and first GP visit to the model, this did not influence the selection nor the coefficients of the predictors. To clarify this we adjusted the following sentence: 'Adjustment for time from baseline to first consultation did not affect the magnitude of the coefficients of the predictors in a substantial way, nor their selection.' (page 9)

The total score for each patient was calculated as the sum of all points for each predictor. Detail the total score ranging from -21 to 301 according to the predictors.

In table 2, we presented the risk score for each predictor. All patients have a combination of these risk score, for example: -21 represents the following patient: 18 years (18), male (-15), healthy activity (-24), no GAD (0), no GP consultation last year (0) ($=18-15-24+0+0=-21$) and 301: 63 years (63), female (0), lack of healthy activity (0), presence of GAD (28), 42 GP consultations last year (210) ($=63+0+0+28+210=301$). To clarify this we added the following information below table 3: 'Note: the risk score was calculated by multiplying each risk score by the predictor value, with the total score ranging from -21 to 301 for all included patients (for example -21 represents the following patient: 18 years (18), male (-15), healthy activity (-24), no GAD (0), no GP consultation last year (0) ($=18-15-24+0+0=-21$) and 301: 63 years (63), female (0), lack of healthy activity (0), presence of GAD (28), 42 GP consultations last year (210) ($=63+0+0+28+210=301$).' (page 23)

4- Results: 24% of participants had a missing value and 3% had missing values for >4 predictors. Detail in table.

In supplemental Table 2, you can see that 648 of all 2,650 participants had a missing value resulting in 24%. To clarify this we added a note below the table: 'Note: 24% (648/2,650) had a missing value'. The missing value analysis showed that 80 (3%) participants had a missing value for more than 4 predictors.

Only five predictive factors out of the 14 included in the analysis remained in the final predictive model. What do the authors attribute this to?

We were also surprised that we did only include five general predictors in our final model. There are two possible not mutually exclusive explanations for this selection. First, although the predictors were described in literature, they are not very strong predictors in our data. Second, the predictors are correlated with each other (e.g. higher BMI is associated with stress full life events and neuroticism). Indeed, there are more significant associations in the univariate compared to the multivariate analysis. We discussed our findings in the implications for research and practice section in the discussion.

Standardize the predictor category. Male or female? Lack of healthy activity or healthy activity? Risk or protection factor?

In our model, we used male and healthy activity, but these predictors showed to decrease the risk for repeated consultations with functional symptoms (see the direction of the odds ratio and coefficient).

In the abstract and result section we mention all the predictor categories that increased the risk and therefore mentioned female and lack of healthy activity. To make this more clear we adjusted the following sentence: 'In the final multivariable model, the following five predictors were selected based on increasing the risk of repeated consultations: higher age, female sex, lack of healthy activity, presence of GAD, and having had more GP consultations in the year before first consulting with functional symptoms (Table 3).' (page 9)

5- Tables and Figure

Table 1. Characteristics of included and excluded patient should be deleted.

We believe that the characteristics of included and excluded patients provides insight in the selection of patients and is illustrative for the reader. Therefore, we did not delete this table, but if the editor finds this table unnecessary it could be deleted.

The Figure 1 is sufficient. The Table 1 should show the predictors characteristics of study population at baseline, overall group and missing (N/%).

We added the % of missing to Table 1.

Table 2. should contain the univariate analysis presented as Supplementary Table 3. Univariable analysis of predictors for frequent attendance with functional symptoms

We moved Supplementary Table 3 to Table 2 in the results section.

Figure 2. Relation between the total risk score and the predicted risk of frequent attendance with functional symptoms as supplementary material.

We moved Figure 2 to the supplementary material.

VERSION 2 – REVIEW

REVIEWER	Frans T Smits Dep of General Practice/Family Medicine Amsterdam University Medical Centres Amsterdam, The Netherlands
REVIEW RETURNED	12-Aug-2020

GENERAL COMMENTS	Review of bmjopen-2020-040730.R1 Developing a clinical prediction rule for repeated consultations with functional symptoms in primary care, a cohort study I thank the authors for their answers on the questions and remarks raised by both reviewers. I can agree with most of their replies and they have certainly bettered their article. Nevertheless, I have some comment: - I cannot agree with their reply on the remark about the used diagnostic terms: functional symptoms and Medically Unexplained Symptoms (MUPS). Indeed, I agree that 'somatic symptom
--

	disorder' is another category, but in my opinion you have to discuss in the introduction or in the discussion ('comparison with other studies') why you choose 'functional symptoms', why you didn't use the term MUPS and why you hardly used, described and discussed the large literature about MUPS. - Also, assuming that a certain percentage of index consultations is a 'provisional symptom diagnosis', you could, in my opinion, over-estimate the validity of your prediction rule. Data in NPCD may give some insight in how many consultations with a functional-symptom code in the end result in an Episode with a non- functional code. Please discuss this issue in the discussion section (limitations). - In the discussion the authors now state: "However, there are other relevant predictors that were not evaluated in Lifelines and were not requested from the NPCD data, such as panic disorder, lack of mastery, medically unexplained physical symptoms episodes, and psychiatric medication (tranquilizers and antidepressants)." All these data (except mastery) are registered in NPCD and you don't mention why you didn't 'requested' these data and combined these with Lifelines or why you couldn't use them. Now you benefit insufficiently of the combination of Lifelines with NPCD. It probably would have bettered the performance of your prediction rule. Please note and explain your motives. - On page 8 you don't mention all candidate predictors based on literature review and expert opinion (see discussion). Please list all possible predictors and the ones you used and the background of this selection. - In the discussion section (comparison with other studies) you better not use 'another study', but list the included studies: The first...The second etc.
--	--

REVIEWER	VERA MARIA VIEIRA PANIZ Universidade do Vale do Rio dos Sinos, Brazil
REVIEW RETURNED	28-Aug-2020

GENERAL COMMENTS	The authors have done a good job and answered most of the comments. The manuscript now provides a better description and presentation of results than previously.
---

VERSION 2 – AUTHOR RESPONSE

Reviewer 1

Review of bmjopen-2020-040730.R1 Developing a clinical prediction rule for repeated consultations for functional symptoms in primary care, a cohort study

I thank the authors for their answers on the questions and remarks raised by both reviewers. I can agree with most of their replies and they have certainly bettered their article.

Nevertheless, I have some comment:

- I cannot agree with their reply on the remark about the used diagnostic terms: functional symptoms and Medically Unexplained Symptoms (MUPS). Indeed, I agree that 'somatic symptom disorder' is another category, but in my opinion you have to discuss in the introduction or in the discussion

(‘comparison with other studies’) why you choose ‘functional symptoms’, why you didn’t use the term MUPS and why you hardly used, described and discussed the large literature about MUPS.

Indeed, we have chosen to use the term “functional symptoms”, which is synonymous for the term medically unexplained physical symptoms (MUPS). We acknowledge that MUPS is more often used in the recent literature compared to functional symptoms and therefore we accommodated your comment as follows. We carefully considered to use MUPS, but after all we changed to the term Functional Somatic Symptoms (FSS) as patients prefer this term over MUPS (Stone et al., BMJ 2002). In addition, FSS is a more neutral term compared to MUPS. We added a sentence in the introduction about the synonymous terms: Functional somatic symptoms (FSS), a synonymous of medically unexplained physical symptoms (MUPS) (page 4).

To our knowledge, we discussed all other studies that developed a clinical prediction rule for persistent FSS using backward or forward selection procedure in the comparison with other studies section. We added the following to the discussion: However, we found three studies that developed models for persistent FSS by combining predictors using a backward or forward selection procedure. We limit our discussion to these three studies that developed a clinical prediction rule.

- Also, assuming that a certain percentage of index consultations is a ‘provisional symptom diagnosis’, you could, in my opinion, over-estimate the validity of your prediction rule. Data in NPCD may give some insight in how many consultations with a functional-symptom code in the end result in an Episode with a non- functional code. Please discuss this issue in the discussion section (limitations).

We agree and we now discussed this issue in the discussion section: A developing underlying somatic disease could be suggested to ultimately explain some of these symptoms, however, a meta-analysis suggested that this risk is very low, reporting only 0.5% new diagnoses in follow-up studies of FSS (Eikelboom et al., Journal of psychosomatic research 2016). (Page 10)

- In the discussion the authors now state: “However, there are other relevant predictors that were not evaluated in Lifelines and were not requested from the NPCD data, such as panic disorder, lack of mastery, medically unexplained physical symptoms episodes, and psychiatric medication (tranquilizers and antidepressants).” All these data (except mastery) are registered in NPCD and you don’t mention why you didn’t ‘requested’ these data and combined these with Lifelines or why you couldn’t use them. Now you benefit insufficiently of the combination of Lifelines with NPCD. It probably would have bettered the performance of your prediction rule. Please note and explain your motives.

In the limitation section of the discussion we added an explanation about this issue: As we did not want to include too many predictors per variable to prevent overfitting, we a priori choose which predictors were most relevant and feasible to use in a primary care setting. By this arbitrary selection, we may have missed relevant predictors (e.g. panic disorder and number of physical symptoms) that could have improved the performance of our prediction rule (Smits et al., Journal of psychosomatic research 2014; Olde Hartman et al., Journal of psychosomatic Research. 2009).

- On page 8 you don’t mention all candidate predictors based on literature review and expert opinion (see discussion). Please list all possible predictors and the ones you used and the background of this selection.

During a meeting with experts we discussed a list of predictors and selected those that were relevant, available, and could be easily used in a primary care setting. Our final list of possible predictors consisted of the 14 predictors mentioned in the method section. This prioritizing could have been arbitrary and therefore we mentioned this as a limitation in the discussion section (see previous comment).

- In the discussion section (comparison with other studies) you better not use 'another study', but list the included studies: The first...The second etc.

We adjusted this accordingly in the discussion section.

VERSION 3 – REVIEW

REVIEWER	Frans T Smits Department of General Practice Amsterdam University Medical Center Amsterdam, The Netherlands
REVIEW RETURNED	25-Nov-2020
GENERAL COMMENTS	I thank the authors for their satisfactory answers on the remarks mentioned by the reviewer. They have certainly again bettered their article and I think it is now ready and fit for publication. I have no further remarks.